# A platform-agnostic deep reinforcement learning framework for effective Sim2Real transfer towards autonomous driving
Dianzhao Li [1,2] ✉ & Ostap Okhrin[1,2]

Autonomous driving presents unique challenges, particularly in transferring agents trained in simulation to real-world environments due to the discrepancies between the two. To address this issue, here we propose a robust Deep Reinforcement Learning (DRL) framework that incorporates platform-dependent perception modules to extract task-relevant information, enabling the training of a lane-following and overtaking agent in simulation. This framework facilitates the efficient transfer of the DRL agent to new simulated environments and the real world with minimal adjustments. We assess the performance of the agent across various driving scenarios in both simulation and the real world, comparing it to human drivers and a proportional-integral-derivative (PID) baseline in simulation. Additionally, we contrast it with other DRL baselines to clarify the rationale behind choosing this framework. Our proposed approach helps bridge the gaps between different platforms and the Simulation to Reality (Sim2Real) gap, allowing the trained agent to perform consistently in both simulation and real-world scenarios, effectively driving the vehicle.

Recently, there has been an encouraging evolution in Deep Reinforcement Learning (DRL) for solving complex tasks[1–9]. Consequently, DRL is increasingly attractive for use in transportation scenarios[10–12], where vehicles must make continuous decisions in a dynamic environment, and behaviors in the real-world traffic can be extremely challenging.

Fully autonomous driving consists of multiple simple driving tasks, for instance, car following, lane following, overtaking, collision avoidance, traffic sign recognition, etc. Simple tasks and their combination need to be tackled to build cornerstones in achieving fully autonomous driving[13–15]. For this purpose, researchers use DRL algorithms to solve various tasks, such as car following[16], lane keeping[17], lane changing[18], overtaking[19], and collision avoidance[20]. Additionally, to improve driving performance, researchers often combine human prior knowledge with DRL algorithms[21,22].

Developing a fully autonomous driving system with DRL algorithms requires training an agent in a simulator and implementing it in real-world scenarios. Since training a DRL agent directly in the real world is impractical due to the enormous number of interactions required to improve policies, simulation plays here a crucial role. Despite the increasing realism of simulators, there are differences between simulation and reality that create a Simulation to Reality (Sim2Real) gap, making the transfer of DRL policies from simulation to the real world challenging[23]. To address this issue, several techniques in different research fields, such as domain randomization, domain adaptation, knowledge distillation, meta-reinforcement learning,

and robust RL have been proposed[9,24–28] in particular to narrow the gap between the simulation and reality. In transportation research, Sim2Real methods already enable the agents to perform tasks such as lane following effectively[29,30] and collision avoidance maneuvers[31–33] in the real world.

The aforementioned works are mainly focused on a single driving task such as lane following, with no related studies addressing overtaking, let alone the combination of lane following, overtaking, and real-world transformation. This research gap motivated us to propose a robust DRL agent training framework for Sim2Real with the capabilities of following the lane and safely overtaking slower vehicles.

The proposed framework shown in Fig. 1 comprises a platform-dependent perception module and a universal DRL control module. The perception module acts as an abstraction layer that addresses the heterogeneity between different platforms and enhances generalization by extracting task-relevant affordances from the traffic state. The affordances relevant to the lane following and overtaking tasks are then produced as the input state for the control module. The latter is the DRL agent that utilizes the aggregated information from the perception module and trains its driving policy within the simulator for different driving scenarios. To this end, we use a Long Short-Term Memory (LSTM) based DRL algorithm that has been proven to be effective for multiple driving tasks, such as lane following and overtaking[34–38]. The DRL agent is first trained on a simple circular map within the Gazebo

[1]Chair of Econometrics and Statistics, esp. in the Transport Sector, Technische Universität Dresden, Dresden, Germany. [2]Center for Scalable Data Analytics and Artificial Intelligence (ScaDS.AI) Dresden/Leipzig, Dresden, Germany. ✉e-mail: dianzhao.li@tu-dresden.de

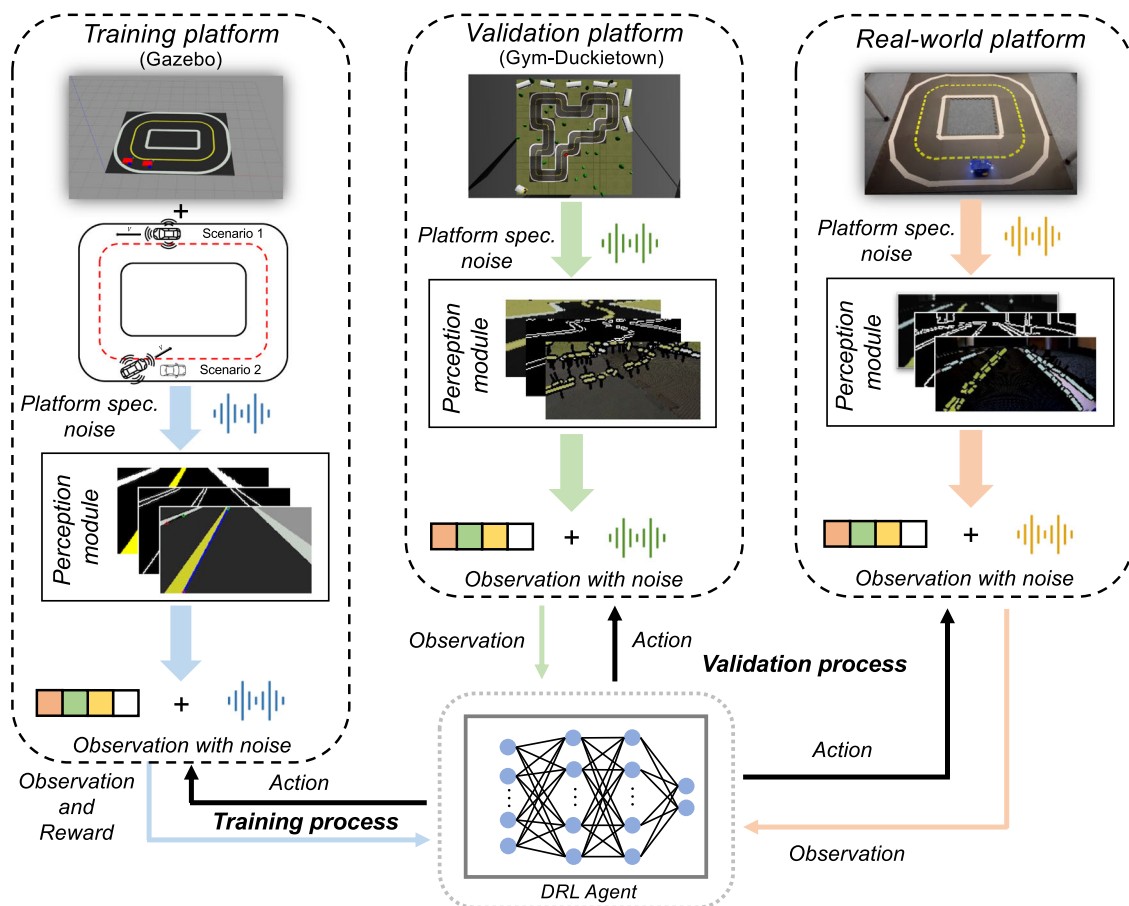

**Fig. 1 | Robust Sim2Real transfer with DRL agent and platform-dependent perception module that separates the agent from the environment.** During the training process, two different driving scenarios for the agent to tackle, are lane following (Scenario 1) and overtaking (Scenario 2) tasks.

simulator[39], then directly transferred for validation to another environment namely the Gym-Duckietown environment[40] and finally tested in the real-world scenario without any additional efforts. Due to the separation of the perception module and the control module, the extracted affordances are utilized by the DRL algorithm, therefore, the generalization of the trained agent is ensured. Summarizing, the main contributions of our work are as follows:

- The training framework with the separation of the perception module and DRL control module is proposed to narrow the reality gap and deal with the visual redundancy. It enables the agent to be transferred into real-world driving scenarios and across multiple simulation environments seamlessly.
- Due to the exceptional generalization property of the proposed framework, the DRL is trained in a simple map in one simulator and transferred to more complex maps within other simulators and even real-world scenarios.
- The developed autonomous vehicle, which employs the proposed DRL algorithm, demonstrates the ability to successfully perform lane following and overtaking maneuvers in both simulated environments and real-world scenarios, showcasing the robustness and versatility of the approach.
- The agent shows superior performance compared to the baselines within the simulations and real-world scenarios. Human baselines are collected within the Gym-Duckietown environment for the lane following task, highlighting the super-human performance of the DRL agent.
- The agent is compared with other DRL baselines and evaluated across various real-world conditions, encompassing differences in lane color,

lane width, and even different platforms. Its consistent and robust performance across diverse scenarios serves as evidence to underscore the effectiveness and rationale behind adopting the proposed agent.

## Results
In this section, we compare the performance of the trained DRL agent with benchmarks in both evaluation and real-world environments and show the results with multiple metrics.

### Performance metrics
Lane following evaluation in simulation: For a quantitative demonstration of the performance, five metrics are used to compare the lane following capabilities in simulation: (1) survival time ($T_s$), the simulation is terminated when the vehicle drives outside of the road, (2) traveled distance ($d_t$) along the right lane of evaluation episodes, (3) lateral deviation ($\delta_l$) from the right lane center, (4) orientation deviation ($\delta_\phi$) from the tangent of the right lane center, and (5) major infractions ($i_m$), which are measured in the time spent outside of the drivable zones. Both deviations $\delta_l$ and $\delta_\phi$ are integrated over time.

We consider the objective of the lane-following task to be driving the vehicle within the boundaries of the right lane with minimal lateral and orientation deviations. In order to provide an exhaustive evaluation and to make a more informative comparison between agents, a final score is used to evaluate the lane-following ability. This score takes into account the performance on each of the individual metrics, weighting them appropriately to provide a comprehensive and fair evaluation of the overall performance of the

agents:

$$\text{Score} = T_s + d_t - \delta_l - \frac{1}{2}\delta_\phi - \frac{3}{2}i_m. \tag{1}$$

The traveled distance and survival time, are set as the primary performance metrics for the lane following task[41]. Furthermore, the penalty is given when the vehicle is driven with deviation from the right lane center or even completely on the left lane. The penalty weights are determined based on the severity of the undesired behaviors. While driving, it is important to avoid the left lane as much as possible. Additionally, the lateral deviation should be kept within a small tolerance. Orientation deviation is less critical but still should be minimized. Worth mentioning is, that function (1) is not used as the reward function for the DRL agent, but purely for comparison and evaluation purposes.

Overtaking evaluation in simulation: In addition to evaluating lane following capability, we also assess the overtaking ability of the trained agent. Thus, we add one more performance metric, the success rate of overtaking maneuvers to the validation process, which is calculated as the ratio between the number of successful overtaking actions and the total number of overtaking maneuvers initiated by the agent. We also retain the other performance metrics to indicate whether the agent can perform lane following under the influence of overtaking behavior.

Evaluation in a real-world environment: For the lane following evaluation in the real-world scenario, only four metrics are selected to evaluate the performance: lateral deviation ($\delta_l$) and orientation deviation ($\delta_\phi$) measured from the perception module, average velocity, and infractions. While the lateral and orientation deviations are not exact measurements, they provide a rough impression of staying within the lane and maintaining the correct orientation. The infraction metric is triggered when the agent veers off the road and requires human intervention to return to the correct path.

## Evaluation maps

During the training phase, we utilize a simple circular map depicted in the training platform of Fig. 1. However, for evaluation purposes, we employ more challenging maps to test the robustness of the trained agent. Specifically, for the Gym-Duckietown environment, we use five different maps, namely Normal 1, Normal 2, Plus track, Zig-Zag, and V track, as illustrated in Fig. 2a, to evaluate the lane following capabilities of the agents[40]. These five maps allow us to test the ability of the agents to maintain lanes in various driving scenarios, including straight lines, curves, and continuous curves. To evaluate the overtaking performance of the agents in simulation, we use only three tracks, namely Normal 1, Normal 2, and Zig-Zag, which are selected due to their relatively straight segments, making them suitable for overtaking maneuvers. Finally, to evaluate the performance in real-world scenarios, we employed a circular map along with five other distinct maps, as depicted in Fig. 3, to evaluate both lane following and overtaking tasks.

## Lane following in simulation

This section presents an evaluation of lane following ability for the trained DRL agent in the Gym-Duckietown environment. We compare the DRL agent with two benchmarks: the classical proportional-integral-derivative (PID) controller[42] and the human driving baseline. In the validation, the PID baseline controller obtains precise information, such as the relative position and orientation of the vehicle to the right lane center directly from the simulator. This information is not available to the DRL agent or human players. The human baseline is driven by human players to perform lane-following maneuvers on five evaluation maps. The DRL agent utilizes processed information from the perception module to generate control commands for the robot, as detailed in PID controller subsection of the Methods section. The PID controller[42], however, has access to the simulator, therefore it can be regarded as a near-optimal controller for the lane following task in the simulation. To assess the robustness of the lane-following capability of the DRL agent, we introduce a feature that scales its action output, allowing for switching between fast and slow driving modes with

different driving preferences. This feature enables us to distinguish between a sporty and calm driver.

The DRL agent and PID baseline are evaluated in 100 episodes across five different maps, with the vehicle being spawned at random positions and orientations before each episode to assess the robustness of their lane-following capability. The median value of their performance across the episodes is used as the evaluation statistic. For the human baseline, however, only the best performance is used for comparison. The evaluation results for the DRL agent, best human baseline, and PID controller are shown in Table 1 and Fig. 2. The exemplary vehicle trajectories from evaluation episodes for each agents are depicted in Fig. 2a. The vehicle trajectories of the slow-mode DRL agent are as smooth as the near-optimal PID baseline controller. For a more quantitative analysis, Table 1, Fig. 2b, and Fig. 2c show that the fast-mode DRL agent achieves the best performance in every evaluation map regarding the final score and traveled distance. Compared with the near-optimal PID baseline based on the traveled distance, namely the velocity, the DRL agent drives almost 50 ~ 70% faster. The DRL agent also outperforms the best human baseline with faster speed and fewer deviations and infractions. Additionally, the slow-mode DRL agent performs even better than the near-optimal PID baseline controller with the same level of speed in terms of lateral and orientation deviation.

To further validate the effectiveness of the proposed DRL agent, we introduce two additional DRL-based baselines: an end-to-end (E2E) DRL agent and a convolutional neural network (CNN) DRL agent. The CNN DRL agent adopts a modular system similar to the proposed DRL agent, using CNN to predict lateral and orientation offsets while employing a DRL agent for control commands. On the other hand, the E2E DRL agent operates as an end-to-end system, directly processing camera images for vehicle control command generation. Detailed descriptions of these baselines are provided in Other baselines subsection of the Methods section. We assess the performance of both DRL baselines using the same validation protocol applied to the other baselines, with results presented in Other baselines subsection of the Methods section.

## Overtaking in simulation

Within this section, we evaluate the overtaking capability of the trained agent in simulation, as there are no existing benchmarks for overtaking, we only access the DRL agent with the proposed metrics. Here only the slow-mode DRL agent is utilized for the overtaking task. Similar to lane following, during the evaluation episodes, the ego vehicle and slower vehicle are spawned at random positions and orientations within three evaluation maps, and the success rates of all episodes are computed. In addition, we also monitor the lane-following performance metrics to assess the lane-following capabilities of agents under the influence of overtaking behaviors. The results are presented in Table 2.

Based on the evaluation results, we observe that the success rates of overtaking maneuvers for all three maps exceeded 90%. The lateral and orientation deviation levels remain consistent with those observed during the lane-following evaluation, which suggests that the overtaking behaviors of the agent did not compromise its lane-following capabilities and that it can maintain an adequate level of control during overtaking maneuvers. However, due to the need for the agent to overtake obstacles and drive on the left lane, the magnitude of infractions observed during the evaluation is greater than that seen during the pure lane following evaluation. Overall, the DRL agent can perform safe overtaking maneuvers when there are obstacles in front and within the detection range. After the overtaking behavior, the agent drives back to the right lane and resumes performing lane following.

## Lane following in real-world environment

For the Sim2Real transformation, the real-world scenarios in MiniCCAM Lab are used to assess the lane following and overtaking abilities of the trained agent in the real world[43]. To assess real-world lane following performance, we compare the proposed DRL agent against the PID baseline, as well as two other DRL baselines: the E2E DRL agent and the CNN DRL agent, all first tested on the robotic vehicles, DB21. The PID baseline

**Fig. 2 | Evaluation results in Gym-Duckietown.**
**a** Illustration of sampled vehicle trajectories for different approaches within five different maps. **b** Boxplots of different performance metrics for different approaches during the 100 evaluation episodes. Boxes represent the first quartile, median, and third quartile of the evaluation results for the respective metrics, while the whiskers extend to the minimum and maximum values. **c** Final scores comparison between the Deep Reinforcement Learning agent and other baselines.

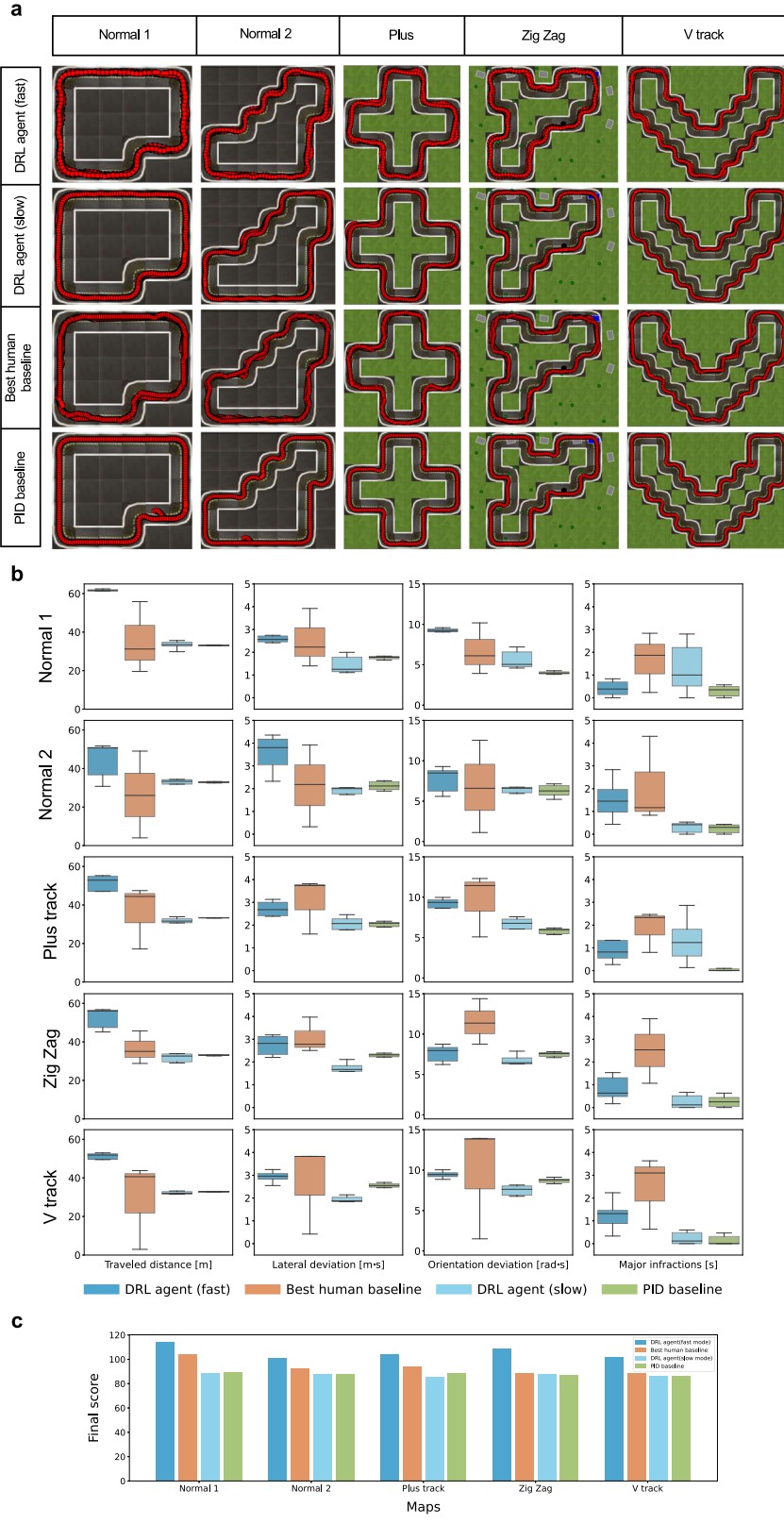

algorithm in this case uses the output information from the perception module to control the vehicle. For the proposed DRL agent, the information from the perception module is used as part of the input state, while the E2E and CNN DRL agents rely directly on image data from cameras to make control decisions. To validate the robustness of the proposed DRL agent, additional tests are conducted using a different type of Duckiebot, DB19,

which is equipped with a Raspberry Pi 3B. The detailed results are included in Supplementary Note 1. For each agent, we perform a 180-second evaluation, capturing video footage for subsequent image processing to analyze performance.

As illustrated by the trajectories in Fig. 3, the proposed DRL agent outperforms the other models during evaluation, exhibiting smooth

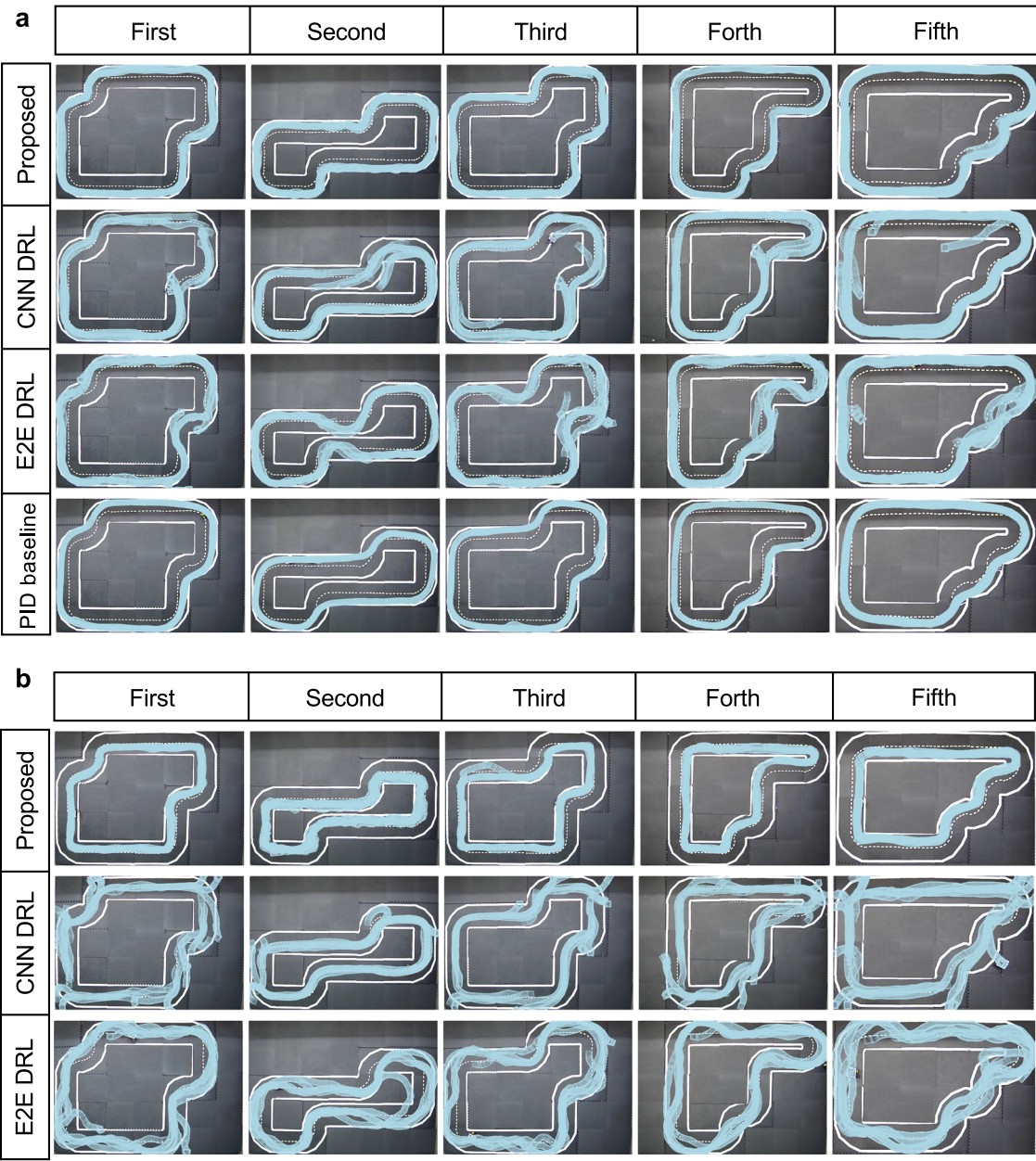

**Fig. 3 | Evaluation results of different approaches for lane following in real-world scenarios. a** Illustration of the vehicle trajectories for the outer ring of five different maps. **b** Trajectories of different approaches running on the inner ring.

turning and consistent performance across both the inner and outer rings of five different maps. Despite exhibiting comparable or even superior performance to the proposed DRL agent in simulation, the two DRL baselines fail when confronted with the complexities of the Sim2-Real gap in real-world scenarios. More specifically, the E2E DRL agent exhibits a more aggressive driving style with larger turning angles. Due to its raw image input features, the E2E DRL agent possesses the capability to maneuver back even from the edge of the left lane, which is an absent feature in the CNN DRL agent. In scenarios where the CNN DRL agent traverses the left lane and approaches its boundary, it struggles to readjust to the right lane. This limitation stems from the infrequency of such instances within the training dataset of the CNN, leading to unpredictable CNN predictions and subsequent infractions during real-world evaluation. This outcome shows the trade-offs between modular and end-to-end systems, signaling the necessity for further refinements in both approaches. As discussed in Other baselines and Sim2Real gap subsections of the Methods section, they also struggle to adapt to other

different real-world variations, leading to a performance drop. The PID baseline, while more stable, tends to drive close to the boundary and frequently cuts curves.

To ensure fairness and provide more quantitative results, we conduct additional experiments using three different vehicles of the DB21, testing on a circular map with both the proposed DRL agent and the PID baseline (Fig. 4a). During the experiments, we logged the lateral deviation and orientation deviation from the perception module, as well as the velocity of the vehicle. The real-world trajectories of the vehicles and the distribution of the lateral and orientation deviation results for different approaches during the evaluation are depicted in Fig. 4. In Supplementary Table S1, the metrics of the two approaches are compared, with the best results being highlighted. For the PID baseline, the average velocity of the vehicle is the maximal possible velocity the baseline algorithm can drive without driving off the road. For each test vehicle, the DRL agent outperforms the PID baseline with respect to the average speed, which is the most important performance metric. Specifically, the DRL agent achieved up to a 65% faster

**Table 1 | Evaluation results for different approaches within different tracks during the lane following evaluation in simulation**

| Maps | Median metric over 100 episodes | Agents | | | |
|---|---|---|---|---|---|
| | | DRL agent[1] | | PID | Human[2] |
| | | slow | fast | baseline | baseline |
| Normal 1 | Final score | 88.50 | **114.52** | 89.18 | 104.01 |
| | Survival Time ($T_s$) [s][3] | 60 | 60 | 60 | 60 |
| | Traveled distance ($d_t$) [m] | 33.75 | **62.27** | 33.17 | 55.82 |
| | Lateral deviation ($\delta_l$) [m · s] | **1.25** | 2.56 | 1.69 | 3.92 |
| | Orientation deviation ($\delta_\phi$) [rad · s] | 5.04 | 9.24 | **3.86** | 10.18 |
| | Major infractions ($i_m$) [s] | 0.99 | 0.38 | **0.25** | 1.87 |
| Normal 2 | Final score | 88.15 | **105.40** | 87.64 | 92.39 |
| | Survival Time ($T_s$) [s] | 60 | 60 | 60 | 60 |
| | Traveled distance ($d_t$) [m] | 34.10 | **53.42** | 32.99 | 49.03 |
| | Lateral deviation ($\delta_l$) [m · s] | 2.99 | 2.54 | **2.02** | 3.92 |
| | Orientation deviation ($\delta_\phi$) [rad · s] | 6.60 | 8.87 | **6.07** | 12.54 |
| | Major infractions ($i_m$) [s] | **0.17** | 0.70 | 0.20 | 4.30 |
| Plus track | Final score | 85.21 | **104.30** | 87.91 | 93.86 |
| | Survival Time ($T_s$) [s] | 60 | 60 | 60 | 60 |
| | Traveled distance ($d_t$) [m] | 32.51 | **52.88** | 33.14 | 47.45 |
| | Lateral deviation ($\delta_l$) [m · s] | **2.07** | 2.68 | 2.10 | 3.74 |
| | Orientation deviation ($\delta_\phi$) [rad · s] | 6.77 | 9.35 | **5.96** | 12.31 |
| | Major infractions ($i_m$) [s] | 1.23 | 0.82 | **0.10** | 2.47 |
| Zig-Zag | Final score | 87.88 | **108.72** | 87.77 | 88.70 |
| | Survival Time ($T_s$) [s] | 60 | 60 | 60 | 60 |
| | Traveled distance ($d_t$) [m] | 32.95 | **56.46** | 33.98 | 45.71 |
| | Lateral deviation ($\delta_l$) [m · s] | **1.67** | 2.82 | 2.24 | 3.97 |
| | Orientation deviation ($\delta_\phi$) [rad · s] | **6.45** | 7.96 | 7.14 | 14.38 |
| | Major infractions ($i_m$) [s] | **0.12** | 0.63 | 0.27 | 3.90 |
| V track | Final score | 86.52 | **102.40** | 85.74 | 88.32 |
| | Survival Time ($T_s$) [s] | 60 | 60 | 60 | 60 |
| | Traveled distance ($d_t$) [m] | 32.38 | **51.77** | 32.94 | 43.76 |
| | Lateral deviation ($\delta_l$) [m · s] | **1.87** | 2.95 | 2.63 | 3.82 |
| | Orientation deviation ($\delta_\phi$) [rad · s] | **7.63** | 10.15 | 9.15 | 13.92 |
| | Major infractions ($i_m$) [s] | 0.12 | 0.90 | **0.0** | 3.10 |

[1]By adjusting the scale of action output, we can switch between fast and slow driving modes.
[2]For the human baseline, we took the best performance.
[3]The maximum evaluation time in one episode is 60s.
*The best performance for each track is highlighted in bold.

**Table 2 | Evaluation results of DRL agent for overtaking behavior on different maps**

| Median metric over 10 episodes | Maps | | |
|---|---|---|---|
| | Normal 1 | Normal 2 | Zig-Zag |
| Success rate[1] | 94.74 % | 94.44 % | 90.91 % |
| Survival Time ($T_s$) [s] | 60 | 60 | 60 |
| Traveled distance ($d_t$) [m] | 21.90 | 21.42 | 22.29 |
| Lateral deviation ($\delta_l$) [m · s] | 1.59 | 2.16 | 2.79 |
| Orientation deviation ($\delta_\phi$) [rad · s] | 5.33 | 7.30 | 8.88 |
| Major infractions ($i_m$) [s] | 5.97 | 5.07 | 2.38 |

[1]The success rate is computed based on all 10 episodes.

range. The same tracks used in the lane following evaluation were employed for this assessment, with a global localization system provided by detecting the Apriltag mounted on top of the vehicles. As there is no baseline controller for overtaking maneuvers of the real vehicle, we only collect the results of the trained DRL agent.

In evaluating overtaking performance, we designed two distinct scenarios for each map. The first scenario involves three static vehicles parked on the road. In this case, the agent must navigate around all three obstacles while maintaining proper lane-following maneuvers before and after each avoidance. The second scenario involves a dynamic overtaking challenge, where a slower vehicle, controlled by a PID baseline, follows the lane while a faster vehicle, under the control of the DRL agent, approaches from behind. Once the slower vehicle is detected within the detection range, the overtaking mode is initiated and the DRL agent executes the overtaking maneuver. Once the overtaking is completed, the agent promptly maneuvers the vehicle back to the right lane and resumes lane following. In addition to the successful overtaking behavior, the DRL agent is also equipped to handle unsatisfactory overtaking situations, where it may drive off the road after passing other vehicles. In such cases, the agent is capable of recovering and guiding the vehicle back on track to resume lane following, demonstrating the robustness of the DRL agent. The trajectories of the proposed agent running evaluations are shown in Fig. 5. The results demonstrate that the agent successfully executes overtaking maneuvers while maintaining effective lane following across various maps.

**Perception module**
To conduct a comprehensive evaluation of the performance of the DRL agent, it is crucial to consider not only the control module but also the perception module used by the agent. The output image obtained during the multi-step image processing procedure is shown in Fig. 6a. However, in the real-world scenario, the true values of the lateral and orientation deviation from the perception module are not available. Thus, we only evaluate the perception module in the Gym-Duckietown environment, as shown in the boxplots in Fig. 6b. Additionally, to assess the performance of the perception module across different maps, we calculate the root mean square error, as presented in Table 3. The root mean square error ranges from 0.046m to 0.067m, which is relatively large given that the width of the road is only 0.23m.

Despite the inaccurate and variable output information of the perception module for the DRL agent, it exhibits superior performance compared to other benchmarks. This suggests that the DRL agent is robust and can handle sub-optimal input states from the perception module.

**Discussion**
Sim2Real is a critical step towards fully autonomous driving using DRL, but discrepancies between different frameworks are hindering this progress. Thus, we propose a training framework that facilitates the transformation of the trained agent between two different platforms, e.g., Sim2Real transfer. The proposed framework includes the perception module with platform-

average speed than the PID baseline controller. Moreover, the DRL agent had only one infraction during the evaluation for all three vehicles. Furthermore, as depicted in Fig. 4, the DRL agent achieved the same or better levels of lateral and orientation deviations compared to the PID controller.

**Overtaking in real-world environment**
Same as in the simulation, the overtaking capability of the trained DRL agent in the real world is also assessed. The vehicle used during the lane following is upgraded with LIDAR equipment for a wider range of distance detection. For the LIDAR equipment, we utilize an RPLIDAR A1M8, which is a low-cost 2D laser scanner solution that can perform 360° scanning within 12m

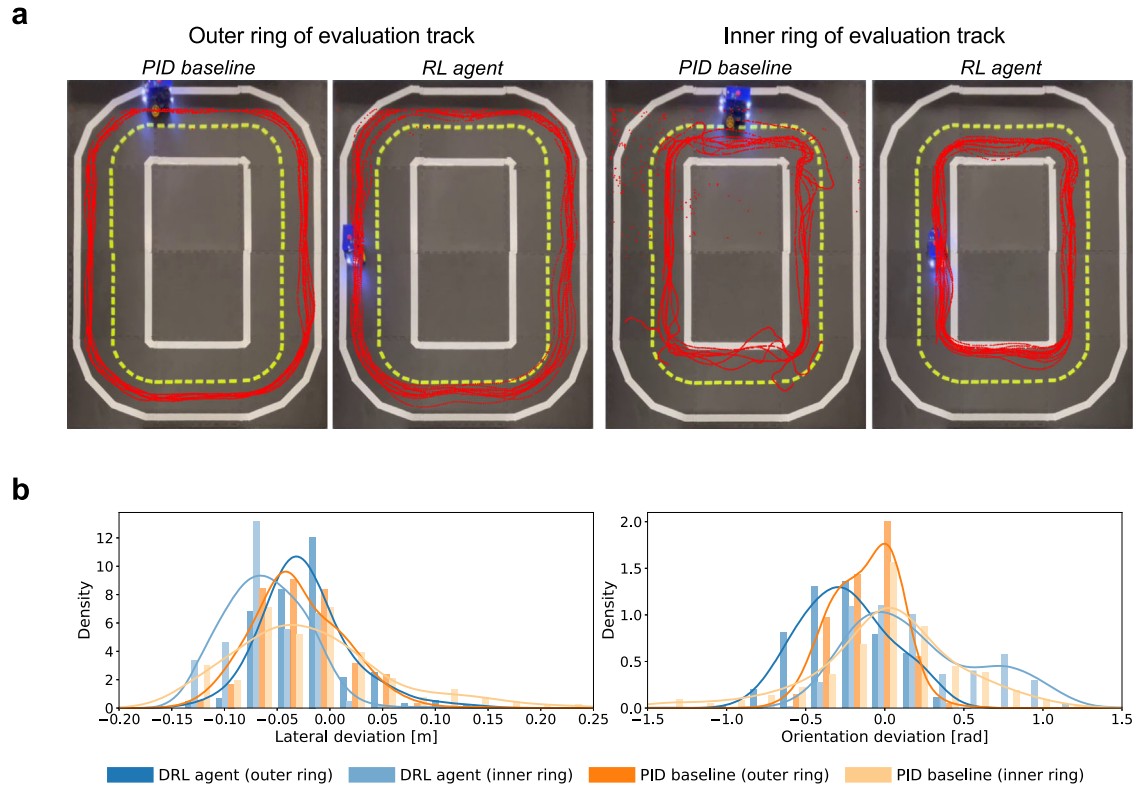

**Fig. 4 | Evaluation results for lane following in real-world scenario for one vehicle, the other two are shown in Supplementary Fig. S2. a** Illustration of the vehicle trajectories for different approaches i.e. Deep Reinforcement Learning (DRL) agent and proportional-integral-derivative (PID) baseline on the inner and outer ring of the real-world track. **b** Distribution histogram and kernel density estimate (KDE) plots of the lateral and orientation deviation for PID baseline and DRL agent during the real-world lane following evaluation.

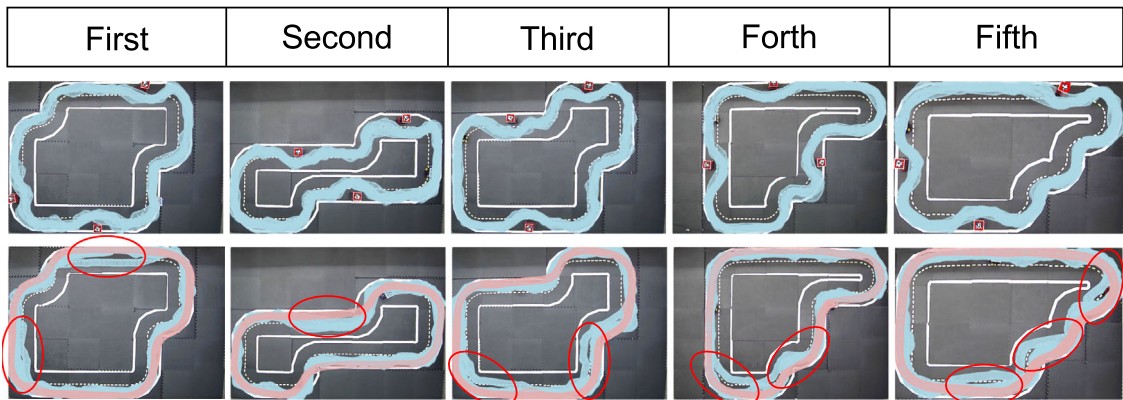

**Fig. 5 | Evaluation results for real-world overtaking scenarios.** The first row depicts the trajectories of the proposed agent as it successfully navigates around three static vehicles (with red rectangles). The second row illustrates the agent overtaking a slower vehicle, shown in light red, with red circles marking the exact positions where the overtaking maneuvers occurred.

dependent parameters and the control module with the DRL algorithm. The platform-dependent parametric perception module improves the generalization of the overall system and simplifies the transformation of the trained DRL agent. It also enables the agent to adapt to different environments and improves its ability to handle variations in input data. For the control module, we employ LSTM-based DRL algorithms, which can effectively handle time-dependent decision-making tasks such as lane following and overtaking behaviors.

To validate our framework, we conducted multiple evaluations in both simulated and real-world environments. First, we validate our trained DRL agent in a Gym-Duckietown environment to perform lane following and overtaking behaviors. Despite the challenges posed by the poor performance

of the perception module discussed in Perception module subsection of the Methods section, the DRL agent still outperforms benchmarks such as PID and trained human players, demonstrating the effectiveness and robustness of our framework. We also evaluate the DRL agent in a real-world environment, where we only modified the parameters in the perception module and applied the trained control module directly. Compared to the PID baseline controller, the trained agent achieved higher speed and fewer infractions during the evaluation, demonstrating its practical applicability. To further illustrate the robustness of the proposed DRL agent, we carefully craft two additional DRL baselines and subject them to validation in various real-world conditions, including different lane colors and lane widths. The proposed DRL agent can maintain its performance under the effect of

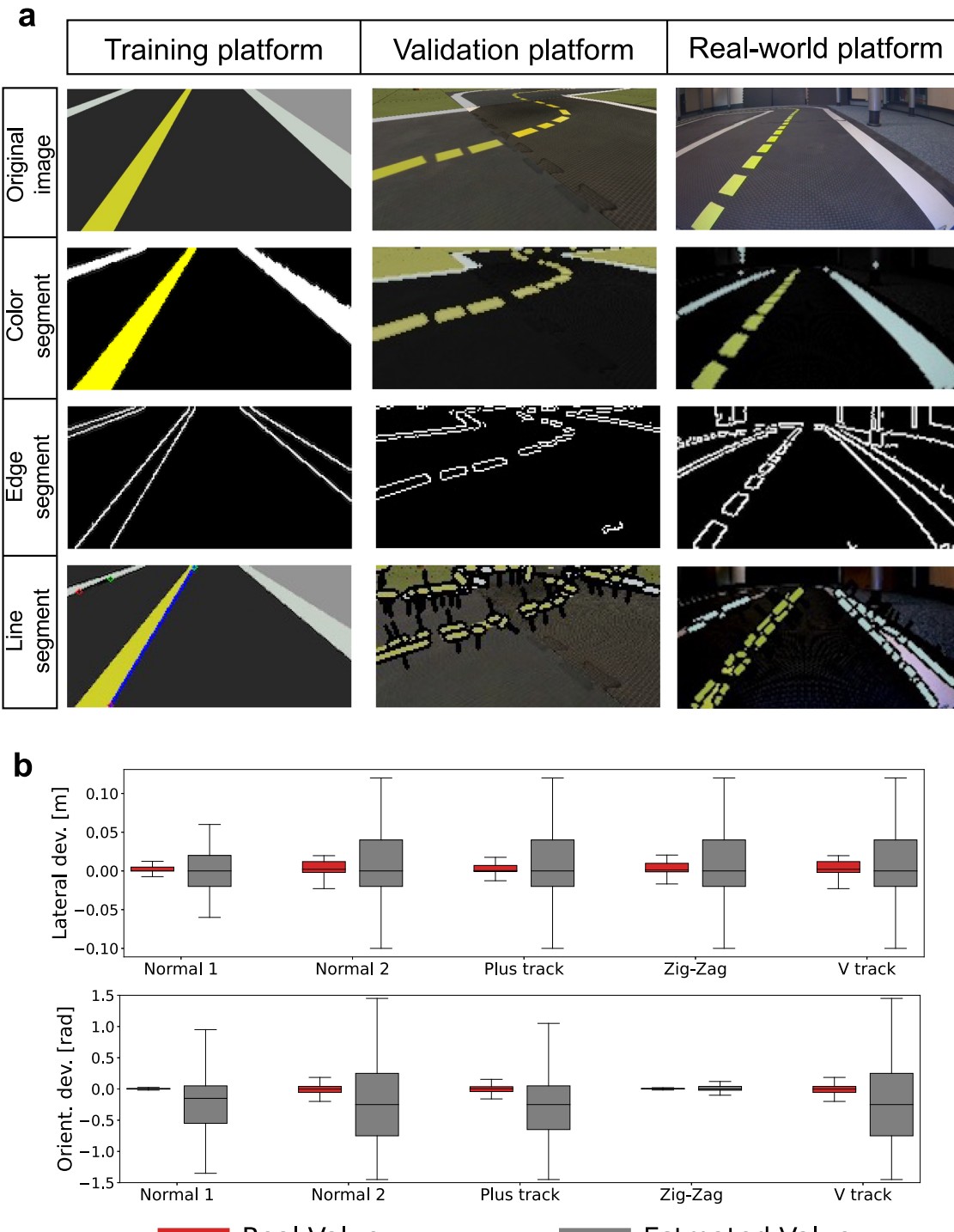

**Fig. 6 | Output results from perception module for different platforms.**
**a** Different outputs during the multi-step perception module, i.e. the image input from the camera, the detected road markings according to the color with Hue-Saturation-Value thresholding, the detected marking edges with canny filter, and final detection marking used to estimate the lateral displacement $d$ and angle offset $\phi$.

**b** The comparison between the real value and the estimated value from the perception module in the validation simulator concerning the lateral and orientation deviation for 10 episodes. Boxes represent the first quartile, median, and third quartile of the deviations across different maps, with whiskers extending to the minimum and maximum values.

different Sim2Real gaps, showing the rationale for choosing such a framework. Additionally, we successfully show the overtaking capability of one faster vehicle in the real-world environment.

While our framework has shown promising results in bridging the gap between different platforms and facilitating Sim2Real transfer, there are still some limitations that call for further improvements. One such limitation is that the framework currently treats vehicles as black boxes and does not

account for their dynamic differences. The control module outputs high-level commands, such as velocity, which may not be suitable for other robotic systems. Another area for improvement lies in addressing noise within the perception and control modules. For instance, incorporating recurrent neural networks, such as LSTMs, into the perception module could help mitigate image transfer latency issues. Additionally, for the overtaking task, the vehicle in front is driven with a constant slow velocity,

**Table 3 | The prediction root mean square error of the perception module regarding the lateral and orientation deviation over different maps**

| Maps | Lateral dev.[m] | Orientation dev. [rad] |
|------|-----------------|------------------------|
| Normal 1 | 0.046 | 0.548 |
| Normal 2 | 0.067 | 0.836 |
| Plus track | 0.050 | 0.672 |
| Zig-Zag | 0.063 | 0.780 |
| Vtrack | 0.066 | 0.833 |

which is not a realistic scenario. A more realistic approach would be to implement a cooperative overtaking behavior where the leader recognizes the overtaking intention of the follower and decides to slow down or maintain a constant velocity to facilitate the overtaking action. This can be achieved using multi-agent reinforcement learning. We plan to explore these improvements in future work, potentially also incorporating advancements observed in the behavior of the E2E DRL agent.

## Methods

### Problem formulation

Achieving fully autonomous driving requires addressing fundamental driving tasks such as car following, lane following, and overtaking. In this work, we focus specifically on lane following and overtaking maneuvers on a two-lane road (see Fig. 1). Our autonomous agent is designed to maintain lane keeping behavior using information collected by sensors when no vehicle is driving in front or no static obstacle is on the road. When a slower-moving vehicle or a static obstacle blocks the path, the agent must safely overtake the obstacle. To achieve a successful transfer of learned behavior from the simulated environment to the evaluation simulator and the real-world environment, the agent must be capable of accounting for any discrepancies between platforms. Moreover, the trained agent must be able to drive the vehicle in the real-world environment without colliding with other vehicles.

To enable the agent to perform these tasks, we equipped the ego vehicle with multiple sensors, including a camera for lane detection and distance sensors for vehicle detection.

### Perception module

In this section, we will describe in detail the multi-step image processing pipeline employed in the perception module, as illustrated in Fig. 6a. The pipeline starts with compensating for the variability in illumination. Initially, the k-means clustering algorithm is used to detect the primary clusters on the road from a subsample of sensor pixels. These clusters are then matched with the expected red, yellow, white, and gray clusters based on prior information, which may slightly vary depending on the environment. An affine transformation is then fitted in the RGB color space between the detected clusters and the color-balanced version of the image to obtain the illumination-corrected image. The line segmentation detection pipeline is then executed using a Canny filter to detect edges[44], and Hue-Saturation-Value colorspace thresholding is employed to detect the expected colors in parallel. The resulting output images are displayed in the second and third columns of Fig. 6a. Next, individual line segments are extracted using a probabilistic Hough transform[45], and the resulting output images are depicted in the fourth column of Fig. 6a. The length and orientation of line segments are used to filter out noise and extract the relevant lines. Reprojection from image space to the world frame is performed based on extrinsic and intrinsic calibration[46]. Using different line segments, a nonlinear non-parametric histogram filter is used to estimate the lateral displacement $d_\delta$ to the right lane center and the angle offset $\theta_\delta$ relative to the center axis. These estimates are utilized later in the control module to facilitate lane following and overtaking behaviors[41].

### PID controller

The PID controller is a classical control theory that has been widely used in the domain of automatic control[42]. Due to its simplicity and effectiveness, it has become a well-established method for controlling a variety of systems, including those in the field of robotics and autonomous vehicles. Its widespread use in industry and academia has led to a deep understanding of its strengths and limitations, making it a valuable benchmark for comparison with newer control methods, such as DRL.

In this study, the proportional-derivative (PD) controller is used as a benchmark for comparison with the performance of the DRL agent in simulation. The PD controller uses information about the relative location and orientation of the robot to the centerline of the right lane, which is available in the simulator. We conduct a series of tests to evaluate the performance of the PD controller using the same information from the perception module as the DRL agent. However, the results showed poor performance of the PD controller, which can be attributed to the uncertainty in the output from the perception module. To achieve more stable control, the velocity of the robot is kept constant, and the orientation $\phi_t$ is controlled using PD control:

$$\phi_t = K_p \theta_{\delta,t} + K_d(\theta_{\delta,t} - \theta_{\delta,t-1}), \tag{2}$$

where $\phi_t$ is the orientation output at time step $t$ and $\theta_{\delta,t}$ is the orientation deviation provided by the simulator. The proportional gain $K_p$ and derivative gain $K_d$ are set as 2.0 and 5.0, respectively.

In real-world lane following scenarios, a PI controller is used as one of the baseline benchmarks. However, precise information on lateral and orientation deviations is unavailable, and instead, features extracted from the perception module are used. The orientation control output in the real world at time step $t$, denoted by $\phi_{t,r}$, is calculated using the following:

$$\phi_{t,r} = K_{p,r}^d d_{\delta,t,r} + K_{i,r}^d \sum_{j=0}^{t} d_{\delta,j,r} + K_{p,r}^\theta \theta_{\delta,t,r}, \tag{3}$$

where $d_{\delta,t,r}$ and $\theta_{\delta,t,r}$ indicate the lateral and orientation deviations from the perception module, respectively. The proportional coefficients $K_{p,r}^d$ and integral coefficients $K_{i,r}^d$ for lateral deviation are set as -6.0 and -5.0, respectively. The proportional coefficients for orientation deviation are denoted by $K_{p,r}^\theta$ and set as -0.3. The parameters in the PID controller for both simulation and the real world are chosen based on multiple experiments[41].

### Human baseline

To compare the lane following performance of the DRL agent with that of human players, we design a keyboard controller to control the robot in the Gym-Duckietown environment and collect data on the performance of human players. For perception, human players observe the same image output from the camera in front of the robot and use the human internal visual perception function to locate its position. We recruit 25 human players, including 16 males and 9 females, with an average age of 31, who are mainly researchers in transportation or computer science, with varying levels of expertise in robotics and autonomous driving.

To ensure fairness and accuracy in the comparison between the DRL agent and human players, each human player is given a minimum of 20 minutes to train using five different maps that are later used for evaluation. This allows the human players to become familiar with the Gym-Duckietown environment and the perception module used in the study. During the evaluation, each human player has six attempts to drive within the evaluation tracks, and only the best performance in terms of the final score (1) is recorded as their final result. This is done to ensure that the human players are given a fair chance to perform at their best, without the pressure of having only one attempt.

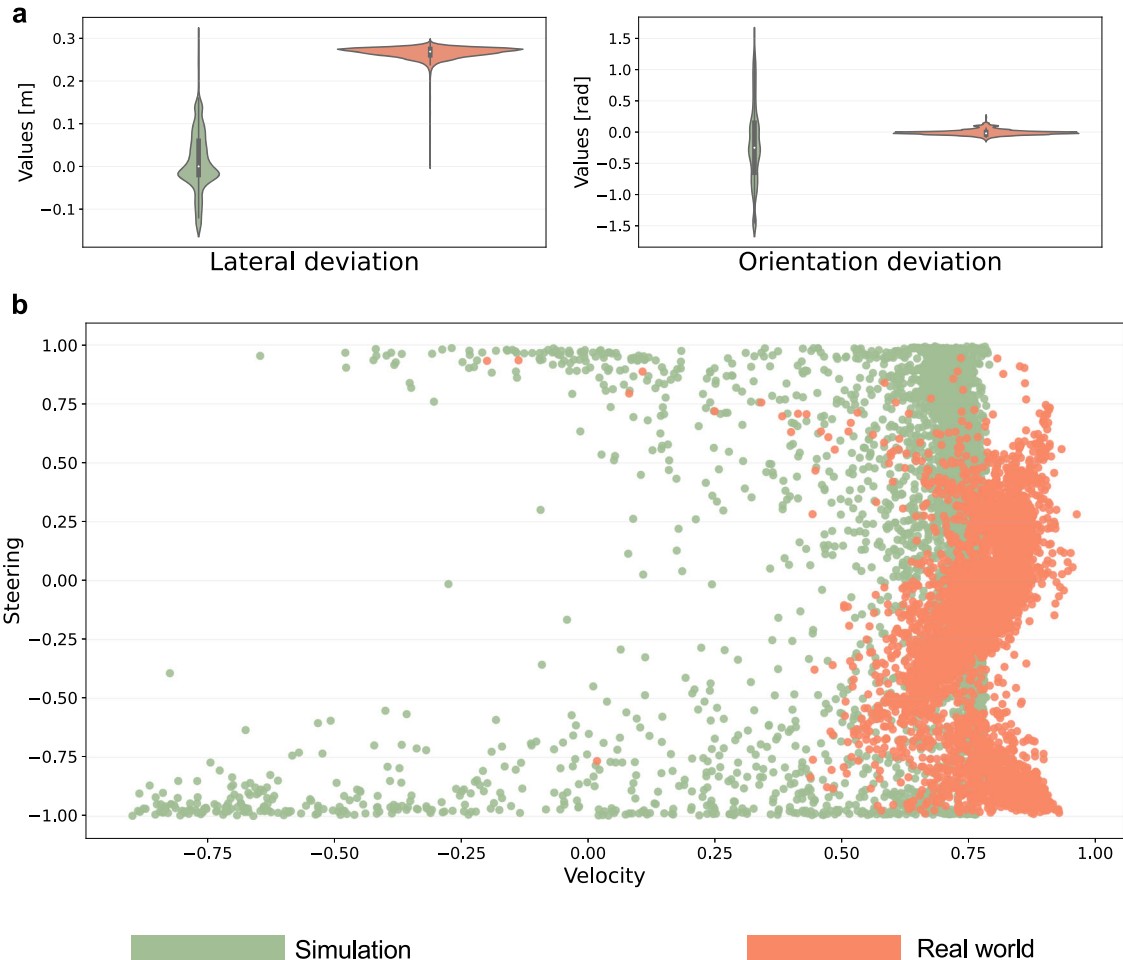

**Fig. 7 | Illustration of part of the observation and action distributions for the Deep Reinforcement Learning (DRL) agent in the simulation and real world. a** is the lateral and orientation deviation outputted by the perception module for 10 episodes, and **b** is the action distribution for the DRL agent in simulation and the real world.

For transparency and further analysis, the overall data of human players, including their scores and trajectories, are collected and available on the project webpage. In addition, a leaderboard is created to display the top-performing human players, allowing for easy comparison with the performance of the DRL agent[47].

### Vector field guidance (VFG) method

The Vector Field Guidance (VFG) method is a widely used approach for solving the path following problem[48]. In this work, we adapt the VFG method to facilitate the lane following task of the DRL agent. Specifically, we calculate the desired course angle command based on the current position of the ego vehicle and the desired lane configuration as

$$\chi^{d} = \chi^{\infty} \cdot \arctan(ke_y) + \chi^{path}, \qquad (4)$$

where $\chi^{\infty}$ and $k$ represent the maximum course angle variation for path following and convergence rate tuning parameters respectively. And $\chi^{path}$ indicates the direction of the target path defined in the inertial reference frame. The cross track error $e_y$ of the vehicle to the target path can be computed as follows

$$e_y = \sin\left\{\chi^{path} - \arctan\left(\frac{y_a - y_{k-1}}{x_a - x_{k-1}}\right)\right\} \cdot d_{w_{k-1}}, \qquad (5)$$

with $d_{w_{k-1}} = \sqrt{(y_a - y_{k-1})^2 + (x_a - x_{k-1})^2}$, where $(x_a, y_a)$ and $(x_{k-1}, y_{k-1})$ are the coordinates of the agent and the closest point on the target path to the position of the agent respectively.

### Control module

This section introduces the setup used to train various DRL agents for the control module, and subsequently compares their performance to obtain an optimal driving policy.

**Observation and action space.** As previously mentioned, our DRL agent uses the lateral displacement $d_\delta$ and angle offset $\theta_\delta$ from the perception module as observation states. Additionally, we employ the VFG method as guidance states to assist the agent in following the optimal trajectory. However, for the overtaking task, the agent needs to be aware of other vehicles in its vicinity. To address this, we include the time-to-collision with other vehicles as a conditional observation state. When other vehicles are detected within the range, this state is activated and works as a regular input state. Otherwise, it is set to zero. Furthermore, we perform input state normalization to speed up the learning process and enable faster convergence.

The overall input state $s_t$ for the DRL agent at time step $t$ is thus defined as

$$s_t = \left(\frac{d_{\delta,t}}{d_w}, \frac{\theta_{\delta,t}}{\pi}, v_t, \frac{|\chi_{yaw,t}|}{2\pi}, e_{y,t}, \rho_l T_c, a_{v,t-1}, a_{\phi,t-1}\right)^\top, \qquad (6)$$

where $d_w$ denotes the lane width, $v_t$ denotes the velocity of the ego vehicle, and $\chi_{yaw,t}$ defines the angle offset between the current heading and the desired heading in the optimal trajectory, $e_{y,t}$ indicates the cross-track error. Here, $\rho_l$ is a binary flag indicating whether there is another vehicle within the detection range $c_d$. Additionally, we use the time-to-collision $T_c = d_{v,t}/v_t$

between the ego vehicle and another vehicle, where $d_{v,t}$ represents the distance between the two vehicles. The action space of our DRL agent consists of the speed command $v_{c,t}$ and steering angle $\phi_t$. Note that the VFG method input states $\chi_{yaw,t}$ and $e_{y,t}$ are only used during the training process. For evaluation, the VFG is not available, we replace $\chi_{yaw,t}$ and $e_{y,t}$ with $\theta_\delta$ and $d_\delta$ from perception module. The hyperparameters used for the input states and reward function are shown in Supplementary Table S2.

**Reward function.** The reward function formulation is a crucial aspect of the RL algorithm as it defines the desired behaviors of the agent. In our case, the objective of the agent is to follow the right lane and overtake slower vehicles when necessary. Therefore, we design the reward function to balance the driving efficiency, lane following, and overtaking capability of the agent:

$$R_t = \begin{cases} -1, & \text{if collide or out of boundary,} \\ w_c R_{c,t} + w_v R_{v,t} + w_\theta R_{\theta,t}, & \text{otherwise .} \end{cases}$$

$$(7)$$

The agent obtains a penalty of -1 when collisions have occurred or if drove off the road. Otherwise, weighted positive rewards are given. The sub-reward functions are shown in Supplementary Table S3. The first term, $R_{c,t}$, guides the agent to learn lane following ability by utilizing the cross-track error $e_{y,t}$ towards the target path, where $k_{r_c}$ is a tunable parameter that controls sensitivity. The second term $R_{v,t}$ devises for driving efficiency, and for overtaking behavior, the agent should leverage the positive velocity reward and collision penalty and obtain suitable overtaking behavior. The third term, $R_{\theta,t}$, penalizes the agent for driving in the opposite direction. As the weights measure the trade-off of different sub-rewards, we conduct many experiments, test different combinations of weights, and select the set of weights that has the best performance. The weights for lane following part $w_c = 0.3$, efficiency or overtaking factor $w_v = 0.6$ and the heading error weights $w_\theta = 0.1$. It is important to note that this reward function is only used during the training process and is not the same as the score (1) used during the evaluation.

**Simulated and real environment**
In this work, Robot Operating System (ROS) and Gazebo simulator are used as our simulation platform to train the agent. ROS is a widely used open-source robotics middleware suite that offers a set of software frameworks for robot development[49]. It enables communication between various sensors on a single robot and multiple robots using a node-to-node communication mechanism. Gazebo is an open-source 3D robotics simulator with real-world physics in high fidelity. It integrates the ODE physics engine, OpenGL rendering, and supports code for sensor simulation and actuator control. Combining both results into a powerful robot simulator with real vehicle dynamics enables simpler Sim2Real transfer. To assess the robustness of the proposed agent across multiple simulated platforms, Gym-Duckietown is utilized as the validation simulator[40]. We used ROS wrappers in the validation simulator to minimize the efforts for the transformation. To enhance generalization during training and assess robustness across different platforms, various types of noise are introduced. During the training phase, Gaussian noise is applied to both images and control outputs to improve resilience. For validation, we utilize the Gym-Duckietown setup, which employs domain randomization to generate input images with varying colors and lighting conditions. In real-world validation, the robustness of the agent is further tested by introducing a range of real-world scenarios, including variations in lane colors, lane widths, lighting conditions, and even different vehicles with distinct dynamics.

For the hardware configuration, we used two Duckiebots, DB21 and DB19, differing primarily in their computation units. DB21 is equipped with an NVIDIA Jetson Nano 2GB, capable of delivering approximately 472 GFLOPS, while DB19 uses a Raspberry Pi 3B, offering around 32 GFLOPS, which results in much lower processing speed and higher transfer latency[33].

Both Duckiebots are equipped with front-facing 160deg FOV cameras, hall effect sensor wheel encoders, Inertial Measurement Units, and time of flight sensors. The RPLIDAR A1M8 LIDAR is used for additional sensing capabilities, but it is only mounted on DB21. Furthermore, we employ three DB21 Duckiebots and repeat the evaluation multiple times to mitigate random outcomes and ensure fairness in our assessments. Considering that manufacturing differences may lead to variations in the dynamics of individual Duckiebots, testing with three DB21 Duckiebots also serves to validate the robustness of the proposed DRL agent.

**Training process**
During the training process, we introduce a surrounding vehicle with low velocity that performs lane following using the PID baseline algorithm. The optimal trajectories for VFG guidance are collected using this vehicle. To further enhance the performance of the ego vehicle in handling both static and dynamic obstacles, the surrounding vehicle will occasionally stop for five seconds and then continue driving. The ego vehicle, which is driven by the DRL agent, is trained to perform overtaking behavior when it is close to the slower surrounding vehicle. When there is no surrounding vehicle in front, the ego vehicle is trained to maintain driving within the right lane. This setup helps to ensure that the ego vehicle learns to handle different traffic scenarios during training.

In this work, TD3, SAC, LSTM-TD3, and LSTM-SAC with the hyperparameters in Supplementary Table S4 are trained[50–53]. Vanilla TD3 and SAC are trained for comparison with LSTM-based TD3 and SAC. The training processes are carried out with an NVIDIA GeForce RTX 3080 GPU and take roughly forty hours to finish one million timesteps of interaction with the simulated environment. The DRL agents are trained for 1.5 million timesteps with 10 different initial seeds and their training performance is evaluated by computing the average test return of 10 evaluation episodes, which is exponentially smoothed. Supplementary Fig. S3 shows the training performance of all agents, where we observe that the two LSTM-based agents converge successfully during the training process. Although the LSTM-SAC agent converges faster and exhibits greater stability than the LSTM-TD3 agent, they achieve similar performance at the end of the training. However, the TD3 and SAC agents fail to learn and obtain approximately zero rewards, even though they use the same reward function as the LSTM-based agents. This result suggests that the recurrent neural networks are essential for the overtaking task in this work. Since only the distance to the surrounding vehicle is used for the overtaking task, the agent needs the past information to avoid confusion with the same distance observation during the overtaking phases.

**Other baselines**
In addition to the PID controller and human baseline, we carefully craft two additional DRL-based baselines for comparison and demonstrate the reason for choosing the proposed framework. One of these is an E2E DRL agent, which directly processes camera images as input and generates vehicle control commands. To optimize computational efficiency, we downsize the input images from $3 \times 480 \times 640$ to $3 \times 32 \times 32$. For training the E2E DRL agent, we employed the same reward function as the proposed DRL agent. This agent is constructed as a CNN comprising two convolutional layers and three fully connected layers, ending with ReLU activations for each layer. Furthermore, we enhance the robustness and generalization of the model by incorporating domain randomization techniques[25] during training, which involved varying lighting conditions, background colors, and introducing noise to the camera images.

As for our second DRL baseline, we adopt a similar two-module structure to our proposed framework, consisting of a perception module and a control module. While the control module remains unchanged, employing an LSTM-SAC agent as per our proposed method. But for the perception module, instead of using multi-step image processing methods, we use a CNN to predict the lateral displacement and the angle offset. To accomplish this, we generate a training dataset comprising 120,000 images with corresponding labels within the Duckietown environment. This

dataset is crafted through a combination of real-world and simulated images, augmented with techniques like noise injection and adjustments to lighting conditions. Subsequently, we devise a CNN architecture tailored for predicting lateral displacement and angle offset, comprising four convolutional layers followed by two dense layers, ending with LeakyReLU activations.

The two DRL baselines are all trained with only lane following tasks and assessed with lane following tasks. We initially evaluate the performance of two DRL baselines in the simulation. As in Supplementary Note 4, we contrast these baselines with our proposed DRL agent with a slow mode. Within the simulation, all three DRL agents exhibit comparable performance across various performance metrics.

## Sim2Real gap

Given the substantial challenges posed by the Sim2Real gap in implementing DRL agents, we aim to provide a quantitative illustration of this gap by focusing on two key aspects. Given that the cornerstone of our framework lies in the perception module utilizing camera images, we pinpoint the Sim2Real disparity within this module through the appearance gap and content gap[54].

The appearance gap consists of visual differences observed between images from two domains: simulation and the real world. It is the pixel-level differences stemming from factors like variations in lighting conditions. To quantify the appearance gap, we employ the Fréchet Inception Distance (FID)[55], a widely-used metric for assessing the quality of images generated by generative adversarial networks or similar image generation models. FID measures the similarity between the distribution of generated images and that of real images in a high-dimensional feature space. The computation of FID begins with a pre-trained Inception v3 model on the ImageNet dataset, excluding its final classification layer but retaining activations from the last pooling layer. Each image is represented by 2048 activation features, capturing high-level semantics. For each image set, the mean and covariance matrix of these feature representations define the distribution of the image set. To compare two image sets, the following equation is employed to compute FID:

$$d^2\{(\boldsymbol{m}, \boldsymbol{C}), (\boldsymbol{m_\omega}, \boldsymbol{c_\omega})\} = \|\boldsymbol{m} - \boldsymbol{m_\omega}\|^2 + \ \mathrm{Tr} \ \{\boldsymbol{C} + \boldsymbol{C_\omega} - 2(\boldsymbol{CC_\omega})^{1/2}\},$$

$$(8)$$

where $(\boldsymbol{m}, \boldsymbol{C})$ and $(\boldsymbol{m_\omega}, \boldsymbol{C_\omega})$ are the means and covariances of the representations from two image sets respectively, Tr is the trace of the matrix. Here, a higher FID value indicates larger discrepancies between two image datasets[55].

Here we compare the camera images captured within the simulation environment and those obtained from the real world. For this, we deploy a trained DRL agent to navigate five distinct maps within the simulation, alongside traversing the inner and outer rings of the real-world environment, to collect images. In total, we collect 6000 images per simulation map and 10000 images from the real world. Subsequently, we compute the FID using (8), resulting in an FID score of 198.82. As noted in[55], FID scores exceeding 150 signify noticeable differences between two sets of images. Thus, our calculated FID score indicates clear disparities between the simulated and real-world Duckietown environments. These disparities constitute the primary factor contributing to the performance degradation of the E2E DRL agent.

In addition to the appearance gap, the content gap defines disparities in the scene-level distributions between two domains. We define this gap through the dissimilarity of the distributions of predicted values generated by the perception module in both the simulated environment and the real world, respectively. To delve into this gap, we employ the trained DRL agent to conduct evaluations and gather outputs from the perception module, specifically the lateral and orientation deviations. Illustrated in Fig. 7a, the distributions of predicted values in the simulation and real world starkly differ, highlighting distinctions in scene-level characteristics between the two domains. Despite the marked differences in the distribution of

predictions from the perception module, our proposed DRL agent demonstrates superior performance both in simulation and the real world compared to baselines. Moreover, as evident from the varying distributions of output actions in simulation and real-world scenarios of the proposed DRL agent in Fig. 7b, our approach exhibits robustness and adaptability across diverse scenarios.

To underscore the resilience of the proposed DRL agent and highlight the impact of the Sim2Real gap, we validated our trained DRL agent alongside other baseline agents across diverse real-world environments. These environments introduced variations such as changes in lane color (DC) and lane width (DW), each posing distinct challenges to Sim2Real transfer. The DC variation, which modifies the lane color from RGB (193, 227, 18) to RGB (230, 198, 0), increases the complexity of Sim2Real transfer. Meanwhile, the DW variation alters the lane width: it expands to 30 cm within the inner ring, facilitating easier navigation, but narrows in the outer ring to just slightly wider than the vehicle (16 cm), making lane-keeping particularly challenging. We conducted comprehensive tests under these varying conditions to assess the robustness of the three DRL agents. These tests included combinations of different lane colors and widths, and importantly, the parameters of the perception module for the proposed DRL agent were not adjusted to accommodate these changes. The evaluation results in Supplementary Note 5 demonstrate how these variations impact Sim2Real performance. The E2E DRL agent showed a consistent performance decline as the Sim2Real gap widened, while the proposed DRL agent maintained robust performance across various real-world variations. Although the CNN DRL and E2E DRL agents initially performed comparably in simulation, they struggled to overcome the Sim2Real gap during real-world evaluations, leading to performance declines and numerous infractions. In contrast, the proposed DRL agent exhibited superior robustness, consistently driving without requiring human intervention. The CNN DRL agent, while experiencing fewer infractions than the E2E DRL agent, underscored the effectiveness of separating perception and control modules for improved real-world performance.

## Fairness versus baseline approaches

Comparing the DRL agent to other baseline approaches may not be entirely fair, given that humans and algorithms approach tasks differently, with different perceptions and decision-making processes. To ensure a fair evaluation, we consider various aspects.

First is the perception method used by different agents. During the evaluation in the Gym-Duckietown environment, we compare the PID baseline and human players with the DRL agent. The PID baseline uses exact information about its position within the lane, which is not available to the other approaches, giving it a significant advantage. However, when validating in real-world scenarios where the PID baseline and DRL agent have the same level of observation, the performance of the PID baseline is inferior to that of the DRL agent. Human players rely on simple observation to determine their position and react accordingly, which is arguably a distinct advantage to DRL agent.

Next, we turn our attention to vehicle control. Both the PID baseline and DRL agent employ the same action space, while human players use a keyboard to manipulate the vehicle's speed, allowing them to constantly adjust its velocity. As a result, there is no clear advantage among the three approaches.

Moving on to driving strategy, we first note that the PID baseline aims to minimize deviations in current velocity, which may be a minor disadvantage when it comes to maximizing the final score. On the other hand, human players are provided with information on how to achieve a higher score through the final score equation, which may incentivize them to take shortcuts and drive at high speeds, a practice we do not encourage. Finally, while the reward function of the DRL agent requires it to strike a balance between deviations and velocity, the weights in the final score equation differ from those in the reward function. Therefore, we do not believe that the DRL agent has an inherent advantage when it comes to driving strategy.

## Data availability

The data including statistical results and trajectories collected from human players, baseline agents, and DRL agent have been made available[56]. The recorded videos of evaluation results in the simulation and real world are also available.

## Code availability

All code to train the DRL agent was implemented in Python using the deep learning framework Pytorch. The code supports the plot within this paper is also available[56]. We have provided all the source code under the MIT license, making it open and accessible to anyone interested in building upon our work.

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

## Acknowledgements
This work was funded by ScaDS.AI (Center for Scalable Data Analytics and Artificial Intelligence) Dresden/Leipzig. The authors would like to extend their gratitude to Meng Wang and Yikai Zeng for their invaluable support in providing the testbed for real-world evaluations and their assistance throughout the evaluations.

## Author contributions
D.L. and O.O. conceived and designed the project. D.L. developed the training platform, prepared the algorithms, and analyzed the results. O.O. interpreted the results. D.L. wrote the paper, and O.O. edited the paper.

## Funding

## Competing interests
The authors declare no competing interests.
