## [Peer Review File · Communications Engineering]

A Platform-Agnostic Deep Reinforcement Learning Framework for Effective Sim2Real Transfer towards Autonomous Driving

Corresponding Author: Mr Dianzhao Li

Version 0:

Reviewer comments:

Reviewer #1

(Remarks to the Author)

This paper proposes DRL framework that can transfer the learned policy in simulation world to real world.

The proposed framework uses "Platform dependent perception module" to reduce the real-to-virtual gap between the simulation world and the real world.

The results show promising results that the proposed framework can work even better than human baseline.

However, there are still several concerns to be addressed

1. the validation platform and the real-world platform are pretty similar to the training platform. Is the good performance because there is actually small real-to-virtual gap? what happens if the agent experience bigger real-to-virtual gap such as change of lane color, lane width, and multiple lanes?
2. The design of the perception module seems critical to the performance of the proposed framework. Is the perception module also learned? or is this something pre-defined?
3. This paper doesn't really have contribution in terms of proposing new DRL algorithm itself.

Reviewer #3

(Remarks to the Author)

This paper proposed a DRL framework that leverages platform dependent perception modules. It was claimed that the proposed framework significantly reduces the Sim2Real gap.

My main concern is, the paper should mainly present how to determine the Platform spec. noise and how to determine the platform specified parameterized perception module. In section 4.2, only multi-step image processing is introduced. Do you mean real world is different from simulation only in the image. It seems that you used a toy vehicle. How about real vehicle on a real road ? Still multi-step image processing ?

The author also wrote "In real-world lane following scenarios, a PI controller is used." This is confusing. should not you use the DRL control?

The authors should compare the simu2real gap in the proposed framework and that not in the framework

Reviewer #4

(Remarks to the Author)

Paper Summary:

The paper proposes a robust Deep Reinforcement Learning (DRL) framework that leverages the perception module and control module to extract task-relevant information. The extracted information is utilized to train a lane-following and overtaking autonomous vehicle agent. The separation of the perception module and control module of the framework facilitates the transfer of the original simulation to new simulated environments and the real-world scenarios with minimal effort. Based on the evaluation results, the proposed framework reduces the gaps between different platforms and Sim2Real gap, enabling the trained agent to achieve similar performance in both simulations and the real-world scenarios compared to the state-of-the-art architectures.

Paper Strengths:

1. The proposed framework is the first work that combines the lane following and the overtaking tasks at the same time. Based on the experiment results, the proposed framework presents effectiveness on both tasks under complex simulation maps and real-world scenarios.
2. The overall architecture of the framework separates the perception module and control module to narrow the reality gap and deal with the visual redundancy. This architecture enables the agent to be transferred into real-world driving scenarios and across multiple simulation environments seamlessly.
3. The performance metrics and the models in the paper are well-designed, which ensure the robustness of the proposed framework. In the evaluation sections, the compared agents used are the PID controller and best human baseline, which are the most powerful current frameworks. The experimental results illustrate that the proposed DRL framework's performance surpasses both the PID controller and the best human baseline, which proves the effectiveness of the proposed framework.
4. The best human baseline is clearly presented in the paper and the procedure the paper used to settle the best human baseline is impartial. Some of the figures are excellent in the paper, for instance, Fig. 2a explicitly illustrates the vehicle trajectories for different approaches, which gives a straight and visual expression on how different approaches actually behave under different maps.

Paper Weaknesses:

1. The paper claims that the separation between the perception and control modules of the proposed framework enables the seamless transfer from simple maps to different simulation environments and real-world scenarios. However, the paper does not provide transfer performance of the previous frameworks that does not separate these two modules. The paper should provide some evidence of the difficulties of the previous frameworks on the scenario transfer. Also, some basic analysis of the transfer difficulties should be supplied.
2. The evaluations illustrate that the proposed framework can be trained in a simple map in one simulator and transferred to more complex maps and real-world scenarios seamlessly. The paper does not contain the performance analysis on the framework which are trained on more complex maps. Will the more complex maps trained framework perform better? If not, what are the important features to make the framework robust and efficient?
3. On page 9, the paper declares that three different vehicles are utilized for the real-world scenario. Could the authors provide more information on the type of the different vehicles? Also, could the authors give a brief explanation of why these types of vehicles are chosen?

Comments:

The paper proposes a DRL framework to tackle the autonomous vehicle's lane-following and overtaking tasks at the same time. The paper is well written and it takes the reader through the design and performance of the proposed framework. The modules and experiments are dexterously designed and the novelty claimed in the paper is solid (i.e., separate the perception and control modules to enable the seamless scenarios transfer). However, some more experiments and analyses should still be added. For instance, the analysis of why the frameworks that do not separate the perception and control modules are difficult to transfer among scenarios should be included, and the performance comparison between simple maps trained and complex maps trained frameworks should be bestowed.

Version 1:

Reviewer comments:

Reviewer #3

(Remarks to the Author)

I am not satisfied with the revised manuscript.

- (1) The authors still did not specify the details of the noise.
- (2) The real world platform is too simple, even much simpler than that in the validation. The authors should use various real world platforms to demonstrate their results.
- (3) As can be seen from Fig.7, although the performance of the proposed DRL is better than the CNN and E2E, but it is still not satisfactory
- (4) the results of overtaking were not shown in any figure like figure 7
- (5) The authors only consider lane following and overtaking, but the title is Autonomous Driving, which is far more than that.

Reviewer #4

(Remarks to the Author)

Thanks for the authors' careful responses to all my questions. The authors present good explanations for all my concerns. I

do not have further questions and my suggestion is "accept". Below are the comments in detail.

First of all, the procedure proposed in the paper to evaluate the effectiveness of the simulation framework is insightful. To the best of my knowledge, this is the first work that proposes this simulation framework's robustness standard. I appreciate the paper's contribution on how to systematically compare the gap between the simulation environments and real-world scenarios in comprehensive aspects.

Further, the paper presents explanations of why the proposed DRL framework is still efficient when relying on simple map input. This is another consolidated contribution of this paper. For autonomous vehicle research today, the data is not too difficult to collect. However, for some other research areas, the data collection is extremely expensive and time-consuming. How to utilize minimum data to train up a powerful framework is the topic that the researchers should focus on.

Finally, the authors provide information on the type of vehicles used in their experiments. Based on our experiences, the manufacturing differences do affect the real-world scenarios' experimental results significantly. The explanations in the authors' rebuttal file make sense to us.

Version 2:

Reviewer comments:

Reviewer #3

(Remarks to the Author)

I have some minor comments concerning response to my comment 3

1. I do not think it is necessary to present result of the Raspberry Pi 3B
2. The authors claimed that "the proposed DRL agent exhibited superior robustness, achieving zero infractions throughout the evaluation". I do not agree. As it is clear from Fig.10, the toy vehicle sometimes is not able to follow the lane.
3. The authors respond that "We believe that there are areas for improvement in our proposed framework,.....", which should be clearly mentioned in the manuscript.

**A Platform-Agnostic Deep Reinforcement Learning Framework for
Effective Sim2Real Transfer in Autonomous Driving
Communications Engineering**

Answers to Reviewer 1

General comments:

First, we gratefully appreciate for the valuable suggestion of this Reviewer, we think that your suggestions do improve the quality of this manuscript. Below, we remind in boldface font your comments and then present our answers in italic.

Major comments:

1. The validation platform and the real-world platform are pretty similar to the training platform. Is the good performance because there is actually small real-to-virtual gap? what happens if the agent experience bigger real-to-virtual gap such as change of lane color, lane width, and multiple lanes?

*Reply: Thank you for pointing this out to us. We highly value your input and have conducted further validations on this matter. In particular, we have identified the Sim2Real gap through two distinct lenses: the **appearance gap** and the **content gap** [1], as elaborated in **Section 4.10**. To quantify the **appearance gap**, we employed the Fréchet Inception Distance (FID) [2], which measures the similarity between the distributions of two image datasets in a high-dimensional feature space. Our analysis outputs a FID score of 198.82, indicating significant differences between the two sets of images. Furthermore, we delved into the Sim2Real gap at the scene level through the **content gap**. Here, we leveraged a trained DRL agent to conduct evaluations and collect outputs from the perception module, including lateral and orientation deviations. Subsequently, we compared the distribution of the lateral and orientation deviations from the perception modules in both the simulation and the real world. As depicted in **Fig.8a**, the distributions of predicted values in the simulation and real world exhibit significant differences. This observation underscores the substantial distinctions in scene-level characteristics between the two domains.*

In addition to identifying the Sim2Real gap, we incorporated additional scenarios as per your suggestion, including variations in lane colors, lane widths, and combinations thereof. These six diverse real-world scenarios provide a spectrum of Sim2Real gaps for evaluation purposes. For comparative analysis, we crafted two alternative DRL-based baselines: an end-to-end (E2E) DRL agent and a convolutional neural network (CNN) DRL agent. The CNN DRL agent adopts a modular system similar to the proposed DRL

agent, using CNN for predicting lateral and orientation offsets while employing a DRL agent for control commands. Conversely, the E2E DRL agent functions as an end-to-end system, directly processing camera images for generating vehicle control commands. Comprehensive details regarding these two DRL baselines are discussed in **Section 4.9**. The reason for designing these two baselines is to showcase the efficacy and rationale of the proposed DRL framework. As delineated in **Section 4.9**, although the CNN DRL agent and E2E DRL agent demonstrate comparable or even superior performance in simulation compared to the proposed DRL framework, real-world evaluations show a stark contrast. Both the CNN DRL and E2E DRL agents experience a significant performance decline, characterized by numerous infractions. In contrast, our proposed DRL agent exhibits superior robustness, maintaining zero infractions throughout the evaluation.

The results above show that the Sim2Real gap is not small and it is hard to handle by two DRL baselines, both of which face the challenge of Sim2Real transfer. In contrast, our proposed DRL framework demonstrates remarkable resilience, consistently delivering stable performance across a spectrum of evaluation scenarios.

2. The design of the perception module seems critical to the performance of the proposed framework. Is the perception module also learned? or is this something pre-defined?

Reply: Thank you for this remark. In our proposed framework, while the perception module remains pre-defined, it exhibits adaptability across various scenarios. To provide a comparative benchmark, we designed a CNN DRL baseline, wherein a CNN replaces the perception module in our framework. As delineated in the results outlined in **Section 4.9**, our proposed DRL framework demonstrates greater robustness when confronted with diverse real-world scenarios, encompassing variations in lane colors and widths. The key distinction lies in the training process: although we created a diverse dataset for training the CNN and employed data augmentation techniques, the dataset still lacks representation of rare scenarios. Consequently, as illustrated in **Fig. 7** and **Table 7**, the performance of the CNN DRL baseline suffers notably when faced with significant Sim2Real gaps, such as narrower lane widths. This decline stems from CNN’s inability to generalize effectively to previously unseen scenarios, resulting in unpredictable predictions and infractions during evaluations.

For comparison purposes, consistent parameters of the pre-defined perception module are employed across the evaluation of various real-world scenarios to test robustness. Although the perception module used in the proposed framework is conventional, it exhibits a higher degree of adaptability and robustness. Notably, when faced with diverse lane widths, including extremely narrow configurations, the perception module adeptly manages to provide suitable predictions to the DRL control module. This ensures consistent performance across a spectrum of scenarios, underscoring the efficacy of our approach.

3. This paper doesn't really have contribution in terms of proposing new DRL algorithm itself.

Reply: *You are absolutely correct in your assessment. Our aim with this manuscript is not to introduce a novel DRL algorithm. Instead, we focus on presenting a new framework designed to train DRL agents effectively, facilitating their seamless transition from simulation to real-world environments. With various evaluations and comparisons, we show that with the proposed framework, a robust DRL agent for lane keeping and overtaking is trained with Sim2Real transfer capabilities.*

**A Platform-Agnostic Deep Reinforcement Learning Framework for
Effective Sim2Real Transfer in Autonomous Driving
Communications Engineering**

Answers to Reviewer 3

General comments:

First, we gratefully appreciate for the valuable suggestion of this Reviewer, we think that your suggestions do improve the quality of this manuscript. Below, we remind in boldface font your comments and then present our answers in italic.

Major comments:

1. The paper should mainly present how to determine the Platform spec. noise and how to determine the platform specified parameterized perception module. In section 4.2, only multi-step image processing is introduced. Do you mean real world is different from simulation only in the image. It seems that you used a toy vehicle. How about real vehicle on a real road? Still multi-step image processing?

*Reply: We appreciate your suggestion and thank you for bringing this to our attention. In our proposed framework, discussed in **Section 4.2**, the perception module is a pre-defined multi-step image processing module. During the predefined phase, noise is not necessary. However, during evaluations, noise is deliberately introduced into both simulation and real-world scenarios to assess the robustness.*

As observed in various vision-based DRL autonomous driving systems [3, 4, 5], the typical workflow begins with training the DRL agent in a simulation environment before transferring it to a real-world vehicle, be it a small-scale or full-size vehicle. Undoubtedly, the real world presents a higher level of complexity compared to simulation, not only in terms of visual information but also in other aspects. Our manuscript, like many others, operates under the assumption that vision-based DRL systems must effectively handle differences between simulated and real-world camera images. While we indeed utilize a small-scale vehicle in our setups, our goal is to introduce a framework that streamlines and provides the Sim2Real transfer process for DRL-based autonomous vehicles with lane following and overtaking tasks. For the real world autonomous driving, two primary methodologies emerge: end-to-end (E2E) and modular systems. E2E systems utilize raw sensor inputs and directly output control commands for the vehicle. Conversely, modular systems adopt a divided approach, separating the perception and control modules, both of which can be implemented using either conventional or machine learning-based methods. For the perception module, two widely used techniques are: multi-step image processing [6, 7] and convolutional neural networks (CNNs) [8, 9, 10].

To provide a more comprehensive analysis and align with the current state-of-the-art real-world setups, we devised two additional DRL-based baselines. The first is an E2E DRL agent, which directly processes raw images as input and generates corresponding control commands as output. The second baseline involves a CNN-based DRL agent, wherein we substitute the multi-step processing module with a CNN tasked with predicting lateral and orientation deviations. To train the CNN, we collected a diverse dataset comprising samples from both simulation and real-world scenarios, augmented using various techniques. The specifics of these two DRL baselines are elaborated upon in **Section 4.9**. Subsequently, we evaluated the performance of these baselines alongside our proposed DRL agent across a spectrum of real-world scenarios, encompassing variations in lane colors, lane widths, and combinations thereof. As depicted in **Fig.7** and summarized in **Table 7**, our findings indicate that the proposed two-module DRL agent maintains consistent performance across diverse real-world conditions, including scenarios with extremely narrow lane widths (16cm).

2. The author also wrote "In real-world lane following scenarios, a PI controller is used." This is confusing. should not you use the DRL control?

Reply: Thank you, we apologize for this confusion. In **Section 4.3**, our focus primarily centers on the discussion of the PI controller, which serves as one of the baseline controllers used to benchmark and compare performance against our proposed DRL agent. We have revised this sentence to ensure clarity and mitigate any potential misunderstanding.

3. The authors should compare the sim2real gap in the proposed framework and that not in the framework

Reply: Thank you for your comment. Unfortunately, the remark is incomplete, but if we interpret your suggestion correctly, you suggest that we should test the DRL agents in real-world scenarios characterized by significant Sim2Real gaps. In response to your suggestion, we have indeed augmented our evaluation with a broader array of real-world scenarios, encompassing variations such as changes in lane color, lane width, and combinations thereof. During the training of the E2E DRL agent and CNN DRL agent, we proactively addressed the challenge of lane color changes. For the E2E DRL agent, we employed domain randomization [11], which involves varying lighting conditions, background colors, and introducing noise to the camera images. Similarly, for the CNN DRL agent, we augmented the training dataset with variations in lighting to account for lane color changes. However, it's important to note that these changes in lane color and lane width were not explicitly encountered during the training of the proposed DRL agent.

After conducting evaluations across six different real-world scenarios, each presenting varying degrees of Sim2Real transfer difficulties, we compared the performance of the two DRL baselines with our proposed DRL agent. As outlined in **Section 4.9**, while the CNN DRL and E2E DRL agents demonstrated comparable performance to the proposed DRL agent in simulation environments, they encountered challenges when confronted

with large Sim2Real gaps or unseen scenarios during training, for example, narrow lane width. This resulted in a noticeable performance drop in real-world environments. In stark contrast, our proposed DRL agent exhibited consistent performance across diverse real-world conditions, even when encountering scenarios that were not encountered during the training phase. This shows the robustness and adaptability of our approach in handling the complexities inherent in real-world driving scenarios.

**A Platform-Agnostic Deep Reinforcement Learning Framework for
Effective Sim2Real Transfer in Autonomous Driving
Communications Engineering**

Answers to Reviewer 4

General comments:

First of all, we would like to thank the Reviewer for carefully reading the manuscript and for the encouraging comments that improved our paper considerably. Below, we remind in boldface font your comments and then present our answers in italic.

Major comments:

1. The paper claims that the separation between the perception and control modules of the proposed framework enables the seamless transfer from simple maps to different simulation environments and real-world scenarios. However, the paper does not provide transfer performance of the previous frameworks that does not separate these two modules. The paper should provide some evidence of the difficulties of the previous frameworks on the scenario transfer. Also, some basic analysis of the transfer difficulties should be supplied.

Reply: *Thank you for pointing this out to us, this is an excellent suggestion that will enhance the quality of our manuscript. In response to your recommendation, we initially identified the Sim2Real gap within our original setups, focusing on two distinct aspects: the **appearance gap** and the **content gap** [1], as detailed in **Section 4.10**. To quantify the **appearance gap**, we utilized the Fréchet Inception Distance (FID) metric [2]. FID measures the similarity between the distributions of two image datasets in a high-dimensional feature space. Our analysis yielded a FID score of 198.82, indicating noticeable differences between the two sets of images. Furthermore, we explored the Sim2Real gap at the scene-level through the **content gap**. For this analysis, we employed a trained DRL agent to conduct evaluations and gather outputs from the perception module, including lateral and orientation deviations. Subsequently, we compared the distribution of these predictions between the simulation and the real world. As illustrated in **Fig.8a**, significant differences were observed in the distributions of predicted values between the simulation and the real world. This finding highlights substantial distinctions in scene-level characteristics between the two domains.*

Following that, we devised two additional DRL-based baselines: an end-to-end (E2E) DRL agent and a convolutional neural network (CNN) DRL agent. The CNN DRL

agent adopts a modular system akin to the proposed DRL agent, employing CNN to predict lateral and orientation offsets while utilizing a DRL agent for control commands. Conversely, the E2E DRL agent operates as an end-to-end system, directly processing camera images to generate vehicle control commands. Moreover, we incorporated various real-world scenarios with variations in lane colors (DC), lane widths (DW), and combinations thereof. A total of six diverse real-world scenarios were included to provide a spectrum of Sim2Real gaps for evaluation, each presenting distinct challenges to the Sim2Real transfer process. The variation in lane color (DC) increases the complexity of the Sim2Real transfer by modifying the lane color. Meanwhile, variations in lane width (DW) within the inner ring, where lane width expands to 30cm, facilitate easier navigation for vehicles within the lane. Conversely, variations in lane width within the outer ring, narrowing the lane width slightly larger than the vehicle width (16cm), make lane-keeping exceptionally challenging, thereby significantly amplifying the difficulties of Sim2Real transfer. Afterwards, the proposed DRL agent and the two DRL baselines were validated across these real-world scenarios, with the results presented in **Section 4.9**. **Fig. 7** and **Table 7** illustrate how varying degrees of Sim2Real gaps significantly impact the performance of the baselines. Particularly noteworthy is the consistent performance decline observed in the E2E DRL agent as the Sim2Real gap widens. In contrast, the proposed DRL agent maintains its performance across various real-world variations, demonstrating robustness and adaptability. These results underscore the challenge posed by the Sim2Real gap in real-world scenarios, not only for an E2E DRL agent but also for the CNN DRL agent, which also experiences performance decline.

2. The evaluations illustrate that the proposed framework can be trained in a simple map in one simulator and transferred to more complex maps and real-world scenarios seamlessly. The paper does not contain the performance analysis on the framework which are trained on more complex maps. Will the more complex maps trained framework perform better? If not, what are the important features to make the framework robust and efficient?

Reply: Thank you for your comment. Training an E2E DRL agent with complex maps is indeed essential and intuitive, given that E2E DRL agents rely on raw sensor data, such as raw images in our setups, as inputs. Training with complex inputs is crucial for enhancing performance, enabling the agent to adeptly handle both easier and more intricate scenarios during evaluation. This principle guided our approach in training the E2E DRL baseline. Specifically, we trained it within the **Zig-Zag** map, which is the most complex map in our dataset, featuring multiple irrelevant objects alongside the roads. Attempts to train the E2E DRL baseline solely with simpler maps, such as **Normal 1** or **Normal 2**, resulted in failed evaluations due to the heightened complexities encountered during testing.

In contrast, the proposed DRL framework adopts a modular architecture, wherein the system is divided into a perception module and a control module. The perception module is tasked with handling variations in input images and uncertainties, generating

lateral and orientation deviations for the control module. Leveraging multi-step image processing methods, the proposed framework ensures that both complex and simple maps yield the same dimension of required information for the DRL control module. Consequently, this framework facilitates the robust performance of the proposed DRL agent across different evaluation scenarios, even encompassing scenarios with unseen or extreme conditions.

3. On page 9, the paper declares that three different vehicles are utilized for the real-world scenario. Could the authors provide more information on the type of the different vehicles? Also, could the authors give a brief explanation of why these types of vehicles are chosen?

Reply: Thank you for your careful review, we apologize for not clearly explaining this. During the evaluation and the comparison between the proposed DRL agent and the PID baselines, to ensure fairness and prevent random results, we use three Duckiebots and repeat the evaluation multiple times. Furthermore, since the dynamics of individual Duckiebots may vary due to manufacturing differences, testing with three Duckiebots also serves to validate the robustness of the proposed DRL agent. The description of the vehicles used for real-world evaluation is provided in **Section 4.7**, where we have clarified that three Duckiebots were employed for the evaluation.

References

- [1] Viraj Uday Prabhu, David Acuna, Rafid Mahmood, Marc T. Law, Yuan-Hong Liao, Judy Hoffman, Sanja Fidler, and James Lucas. Bridging the sim2real gap with CARE: Supervised detection adaptation with conditional alignment and reweighting. *Transactions on Machine Learning Research*, 2023.
- [2] Martin Heusel, Hubert Ramsauer, Thomas Unterthiner, Bernhard Nessler, and Sepp Hochreiter. Gans trained by a two time-scale update rule converge to a local nash equilibrium. *Advances in neural information processing systems*, 30, 2017.
- [3] Xinlei Pan, Yurong You, Ziyang Wang, and Cewu Lu. Virtual to real reinforcement learning for autonomous driving. *arXiv preprint arXiv:1704.03952*, 2017.
- [4] Maximilian Jaritz, Raoul De Charette, Marin Toromanoff, Etienne Perot, and Fawzi Nashashibi. End-to-end race driving with deep reinforcement learning. In *2018 IEEE international conference on robotics and automation (ICRA)*, pages 2070–2075. IEEE, 2018.
- [5] Shuo Feng, Haowei Sun, Xintao Yan, Haojie Zhu, Zhengxia Zou, Shengyin Shen, and Henry X Liu. Dense reinforcement learning for safety validation of autonomous vehicles. *Nature*, 615(7953):620–627, 2023.

- [6] Sandipann P Narote, Pradnya N Bhujbal, Abhilasha S Narote, and Dhiraj M Dhane. A review of recent advances in lane detection and departure warning system. *Pattern Recognition*, 73:216–234, 2018.
- [7] Christopher Rose, Jordan Britt, John Allen, and David Bevly. An integrated vehicle navigation system utilizing lane-detection and lateral position estimation systems in difficult environments for gps. *IEEE Transactions on Intelligent Transportation Systems*, 15(6):2615–2629, 2014.
- [8] Noor Jannah Zakaria, Mohd Ibrahim Shapiai, Rasli Abd Ghani, Mohd Najib Mohd Yassin, Mohd Zamri Ibrahim, and Nurbaiti Wahid. Lane detection in autonomous vehicles: A systematic review. *IEEE access*, 11:3729–3765, 2023.
- [9] Alexandru Gurchian, Tejaswi Koduri, Smita V Bailur, Kyle J Carey, and Vidya N Murali. Deeplanes: End-to-end lane position estimation using deep neural networks. In *Proceedings of the IEEE conference on computer vision and pattern recognition workshops*, pages 38–45, 2016.
- [10] Youcheng Zhang, Zongqing Lu, Xuechen Zhang, Jing-Hao Xue, and Qingmin Liao. Deep learning in lane marking detection: A survey. *IEEE Transactions on Intelligent Transportation Systems*, 23(7):5976–5992, 2021.
- [11] Josh Tobin, Rachel Fong, Alex Ray, Jonas Schneider, Wojciech Zaremba, and Pieter Abbeel. Domain randomization for transferring deep neural networks from simulation to the real world, 2017. In *IEEE/RSJ international conference on intelligent robots and systems (IROS) 23-30 (IEEE, 2017)*.

**A Platform-Agnostic Deep Reinforcement Learning Framework for
Effective Sim2Real Transfer in Autonomous Driving
Communications Engineering**

Answers to Reviewer 3

General comments:

First, we gratefully appreciate for the valuable suggestion of this Reviewer, we think that your suggestions do improve the quality of this manuscript. Below, we remind in boldface font your comments and then present our answers in blue.

Major comments:

1. The authors still did not specify the details of the noise.

Reply: Thank you for your comment. We apologize for not addressing this matter in our previous revision. In the current revision, we have clarified the noise handling for each platform. Specifically, in Section 4.7, we discuss how noise is applied across different platforms. During the training phase, Gaussian noise is added to both images and control outputs to enhance generalization. For the validation platform, domain randomization within Gym-Duckietown is utilized to generate input images with varying colors and lighting conditions. In real-world validation, the agent's robustness is further evaluated by introducing diverse real-world scenarios, including variations in lane colors, lane widths, lighting conditions, and even different vehicles with distinct dynamics, see Section 4.10.

2. The real world platform is too simple, even much simpler than that in the validation. The authors should use various real world platforms to demonstrate their results.

Reply: Thank you for your comment. According to your suggestion, as detailed in Sections 2.5 and 2.6, we expanded our evaluation by incorporating five additional real-world maps to assess the performance of various approaches, including the proposed DRL agent, the PID baseline, E2E DRL, and the CNN DRL agent for lane-following capabilities. For overtaking assessment, we tested the proposed agent in two different scenarios: one with static obstacles and another with moving obstacles. The results demonstrate that the proposed agent consistently delivers robust performance across these diverse scenarios.

3. As can be seen from Fig.7, although the performance of the proposed DRL is better than the CNN and E2E, but it is still not satisfactory.

Reply: Thank you for your feedback. We understand and acknowledge your concerns regarding the results. However, in this manuscript, our goal is to propose a platform-agnostic framework that is versatile enough to be used across different platforms and adaptable to various real-world robotic vehicles. To achieve this, we not only validated our proposed agent in diverse map environments but also tested it on different robotic vehicles, including one with significantly lower computational power (Raspberry Pi 3B) compared to the Jetson Nano.

As discussed in Section 4.7, the Raspberry Pi 3B offers approximately 32 GFLOPS, while the Jetson Nano 2GB provides around 472 GFLOPS, highlighting the substantial disparity in computational capabilities. Additionally, the latency of image transfer on the Raspberry Pi 3B is an important factor to consider [1]. Based on our approximations within our network setup, the image transfer latency is approximately 0.2 seconds. Given an average speed of 0.4 m/s, this translates to the vehicle moving 8 cm forward during the latency period. This is particularly critical when navigating sharp curves, as illustrated in Fig.3. The vehicle equipped with the Raspberry Pi 3B (denoted in light red) has difficulty handling sharp turns. Despite this, it successfully navigates some curves during the evaluation and maintains acceptable lane following behavior. Specifically, the proposed agent effectively guides the vehicle back to the right lane after encountering poor driving performance on curves due to latency issues. This demonstrates that, despite the platform’s limitations, the proposed framework can effectively mitigate the drawbacks imposed by the platform. In comparison, other baselines running with the Raspberry Pi 3B failed to complete straight paths and a single curve.

We believe that there are areas for improvement in our proposed framework, particularly in addressing the two sources of noise within the perception and control modules. For example, incorporating recurrent neural networks, such as LSTM, into the perception module could help mitigate image transfer latency issues. Although enhancing these aspects could improve overall performance, it may also impact the framework’s generalization capabilities, which is not within the scope of this manuscript. We plan to explore these improvements in future work, potentially also incorporating advancements observed in the behavior of the E2E DRL agent.

4. the results of overtaking were not shown in any figure like figure 7.

Reply: Thank you for your observation. As illustrated in Fig.5, we have now included the trajectories from the real-world overtaking assessments, covering two different scenarios on each map: one with static obstacles and another with a dynamic vehicle.

5. The authors only consider lane following and overtaking, but the title is Autonomous Driving, which is far more than that.

Reply: Thank you for your suggestion. We have revised the title to *A Platform-Agnostic*

*Deep Reinforcement Learning Framework for Effective Sim2Real Transfer **towards** Autonomous Driving* to prevent any further confusion.

References

- [1] Eric Gamess and Sergio Hernandez. Performance evaluation of different raspberry pi models for a broad spectrum of interests. *International Journal of Advanced computer science and applications*, 13(2), 2022.

COMMS-23-0187A
A Platform-Agnostic Deep Reinforcement Learning Framework for
Effective Sim2Real Transfer in Autonomous Driving
Communications Engineering

Answers to Reviewer 4

General comments:

We are grateful for your time and attention to our manuscript. Your comments have greatly improved its quality, and we sincerely appreciate your recommendation.

**A Platform-Agnostic Deep Reinforcement Learning Framework for
Effective Sim2Real Transfer towards Autonomous Driving
Communications Engineering**

Answers to Reviewer 3

General comments:

First, we would like to express our sincere appreciation to the Reviewer for the valuable suggestions, which we believe have significantly enhanced the quality of this manuscript. Below, we have highlighted your comments in bold and provided our responses in blue.

Major comments:

1. I do not think it is necessary to present result of the Raspberry Pi 3B.

Reply: Thank you for your suggestion. We included the Raspberry Pi 3B in our discussion to highlight the limitations of the proposed framework. Specifically, the Raspberry Pi 3B demonstrates that the framework struggles with managing image transfer latency, which is an area we aim to address in future work. We agree that featuring the Raspberry Pi 3B in the main paper's evaluation may not be appropriate. Therefore, we have moved the evaluation results and related discussion to the Supplementary Information.

2. The authors claimed that "the proposed DRL agent exhibited superior robustness, achieving zero infractions throughout the evaluation". I do not agree. As it is clear from Fig.10, the toy vehicle sometimes is not able to follow the lane.

Reply: Thank you. Sorry for this confusion, for infractions, as we defined in Section **Performance metrics**, means the agent veers off the road and requires human intervention to return to the correct path. We see that the proposed DRL agent can drive without human intervention in different real-world conditions. To avoid any confusion, we have revised the sentence as follows: the proposed DRL agent exhibited superior robustness, consistently driving without requiring human intervention.

3. The authors respond that "We believe that there are areas for improvement in our proposed framework,.....", which should be clearly mentioned

in the manuscript.

Reply: Thank you for your comment. We have added these possible improvements in the Discussion Section.